# Reward-Guided Trajectory Distillation for Accelerated Diffusion-Based Video Generation

## Abstract

Recent advancements in video generation models have achieved remarkable quality but often suffer from slow inference due to the iterative denoising processes required by diffusion models. In this paper, we propose a novel distillation pipeline that leverages a reward model to improve the performance of the video generation model. Specifically, our approach distills the 50-step diffusion model into a few-step video generation model through matching the trajectory distribution. Furthermore, we integrate a carefully designed reward model into the training framework. This additional guidance not only mitigates the influence of redundant or uninformative data points during distillation but also enhances the overall generation quality. By optimizing the reward mechanism, the reward model provides fine-grained feedback on semantic consistency, visual fidelity, and temporal coherence. Extensive experiments demonstrate that our method achieves substantial acceleration in video generation.

## 1 Introduction

Video generation has emerged as a pivotal technology in numerous domains, including entertainment, film production, gaming, and personalized media creation. The ability to synthesize high-quality videos that accurately mimic real-world scenes is crucial not only for creative applications but also for practical use cases where realistic temporal dynamics are essential. To this end, recent advances in diffusion models have already proven successful in high-quality video synthesis Blattmann et al. (2023); Brooks et al. (2024); Hong et al. (2022); Yang et al. (2024); Kong et al. (2024); Ma et al. (2024); Wang et al. (2025a); Lin et al. (2024a); Zheng et al. (2024b); Peng et al. (2025). Diffusion models generate outputs by iteratively refining random noise through a denoising process. While this mechanism has led to the production of visually appealing content, the inherent need for a large number of inference steps poses significant challenges. Specifically, the iterative nature of these models results in slow generation speeds and substantial computational resource demands. The temporal dimension of video data further exacerbates these issues, as maintaining both spatial and temporal coherence becomes increasingly complex.

Recent studies have attempted to alleviate these challenges by leveraging knowledge distillation techniques. These efforts aim to condense the lengthy denoising trajectories of diffusion models into fewer, more efficient steps. There have existed several distillation methods in the image generation diffusion model Berthelot et al. (2023); Luo et al. (2024); Lin et al. (2024b); Luo et al. (2025); Frans et al. (2024); Salimans & Ho (2022); Sauer et al. (2024b;a); Wang et al. (2022b); Xu et al. (2024b); Yin et al. (2024b;a); Yan et al. (2024). However, applying such techniques to video generation is still to be explored. The distillation process often suffers from redundant intermediate data points and mismatches between the training and distillation datasets. Additionally, misalignment in the noise distributions during distillation can lead to suboptimal guidance for the student model, resulting in diminished semantic consistency and degraded temporal coherence in the final generated videos.

In this work, we propose an innovative pipeline that addresses these limitations through the integration of diffusion trajectory distillation with a reward model. Our overall approach accelerates the video generation process by distilling the diffusion trajectories of a pre-trained teacher model into a student model capable of generating high-fidelity videos in substantially fewer sampling steps. Concurrently,

we incorporate a carefully designed reward model that evaluates the generated videos based on semantic consistency, visual fidelity, and temporal coherence. By optimizing the reward mechanism, our method effectively enhances the quality of the generated videos while filtering out redundant or uninformative data points during the distillation process.

To summarize, our contributions are as follows:

1. We propose a distillation pipeline incorporating the reward model and GAN loss to enhance the overall efficiency and quality of the video generation process.

2. We incorporate a reward model into our training pipeline that provides fine-grained feedback on semantic consistency, visual fidelity, and temporal coherence, optimizing the generation quality.

3. Extensive experiments demonstrate that our method not only accelerates video generation but also maintains, and in some cases improves, the output quality compared to existing approaches.

## 2 RELATED WORKS

### 2.1 DIFFUSION DISTILLATION

First, we would like to introduce some recent image diffusion distillation works Berthelot et al. (2023); Frans et al. (2024); Lin et al. (2024b); Luo et al. (2024); Salimans & Ho (2022); Sauer et al. (2024a); Wang et al. (2022b); Xu et al. (2024b); Yan et al. (2024); Yin et al. (2024a); Luo et al. (2025); Song et al. (2023); Wang et al. (2024a); Meng et al. (2023); Heek et al. (2024); Luhman & Luhman (2021); Ren et al. (2024); Xu et al. (2024a); Zheng et al. (2024a); Gu et al. (2023); Yin et al. (2024b); Sauer et al. (2024b); Zhou et al. (2024); Luo et al. (2023); Lu & Song (2024); Chen et al. (2025). The progressive distillation Salimans & Ho (2022); Berthelot et al. (2023) tries to progressively distill the long inference steps from the teacher model to a shorter student model. Wang et al. (2022b); Sauer et al. (2024b;a); Lin et al. (2024b) utilizes the GAN discriminator to train a faster diffusion generator directly, and they add adversarial regularizers to alleviate over-smoothness in distilled students. Some works focus on trajectory based distillation. Luo et al. (2025); Zheng et al. (2024a) explicitly match the distribution of the whole sampling path. PerFlow Yan et al. (2024) and Hyper-SD Ren et al. (2024) segment the trajectory into locally invertible flows, making the accelerator plug-and-play. Some works focus on training with a distribution matching loss to achieve score based distillation, such as Luo et al. (2024); Sauer et al. (2024b); Yin et al. (2024a;b), which aligns the distribution of real and fake scores obtained from the diffusion model. The consistency model Song et al. (2023) also focuses on the one-step or few-step inference by enforcing a consistency constraint between noisy and clean distributions. The latent consistency model Luo et al. (2023) first applies the consistency model on the latent space. To reduce the discrete error of the ODE solver, the phased consistency model Wang et al. (2024a) divides the inference process into multiple sub-consistency stages, and the continuous-time consistency model (sCM) Lu & Song (2024) directly refines the loss with the continuous-time formulation. SANA-Sprint Chen et al. (2025) successfully applies the sCM on the SANA models, which can achieve an impressive inference latency of only 0.1s.

In the realm of video diffusion models, several approaches have been developed to accelerate the video generation process. For instance, Animatediff-lightning Lin & Yang (2024) utilizes the progressive adversarial diffusion distillation to support the few-step generation. T2V-Turbo Li et al. (2024a) and T2V-Turbo-V2 Li et al. (2024c) integrate a consistency distillation loss with a reward model to improve generation efficiency. APT Lin et al. (2025) leverages adversarial post-training to achieve one-step video generation. AccVideo Zhang et al. (2025) introduces a trajectory-based distillation strategy and utilizes a high-quality synthetic dataset, ultimately generating 720p videos in just 5 seconds.

### 2.2 REINFORCEMENT LEARNING IN VIDEO GENERATION MODEL

The integration of reinforcement learning techniques into text-to-video generation has fundamentally transformed the field, shifting focus from likelihood-based optimization toward direct alignment with human preferences and perceptual quality metrics. Key methodological advances including DDPO, GRPO, and DPO variants have established robust frameworks for training high-quality

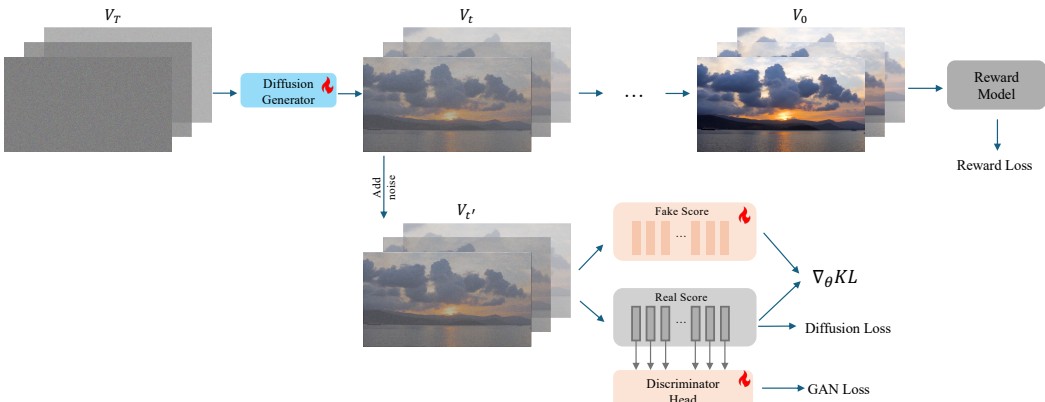

Figure 1: Overview of our reward-guided trajectory distillation pipeline. The student generator $G_\theta$ produces samples at timesteps $t_i$, which are diffused to intermediate timesteps $x_\tau$ by adding noise. Three components jointly optimize the generator: (1) **Trajectory distillation** matches the denoising scores between the student's fake score model and the teacher's real scores; (2) **GAN loss** uses a discriminator head to distinguish real from generated samples, ensuring realistic outputs; (3) **Reward models** evaluate the final output $\hat{x}_0$ for image quality, text alignment, and video coherence. The combined gradient $\nabla_\theta$ enables high-quality video generation in just 4 inference steps.

video generation models that achieve 15-50% performance improvements over traditional supervised approaches. The field's rapid maturation is evidenced by the development of comprehensive evaluation frameworks like VBench and VideoScore, the emergence of physics-aware generation systems, and the transition from experimental approaches to production-ready systems capable of real-time interaction. Li et al. (2024b) Major industry players have adopted hybrid approaches combining the generative capabilities of diffusion models with RL-based alignment techniques for safety, quality control, and user preference optimization.

Future developments are likely to focus on unified multi-modal frameworks, improved computational efficiency, advanced evaluation methodologies for long-form content, and interactive generation systems that enable real-time user control. Liang et al. (2025) The convergence of foundation model capabilities with sophisticated preference learning represents a promising direction toward more general and capable video generation systems that can adapt to diverse user requirements and creative applications across entertainment, education, and content creation industries. Our work builds on this trend by being one of the first to integrate a multi-faceted reward system, combining both image-level and video-level feedback, directly into a distillation pipeline to guide the student model towards producing videos that are not only fast to generate but also semantically correct, aesthetically pleasing, and temporally consistent.

## 3 METHOD

We would like to introduce our distillation pipeline from several aspects in this section. Firstly, we will provide a brief overview of the diffusion model with flow matching in Sec 3.1. And we will talk about the main part of the trajectory diffusion distillation in Sec 3.2.

### 3.1 PRELIMINARIES

The diffusion model (DM) defines a diffusion process that gradually destroys an observed datapoint $x_0 \backsim p_{data}$ over time $t$, by mixing the data with Gaussian noise. We can get $x_t = \alpha_t x_0 + \sigma_t z$, where $t \in [0, T]$, $z \backsim \mathcal{N}(0, I)$. $t$ denotes the timesteps and the $z$ is a standard Gaussian noise. The goal of a diffusion network $\epsilon_\theta$ is to predict a noise, which can be trained by the loss $\mathbb{E}_{x_o, z, t}[||\epsilon_\theta(x_t, t) - z||^2]$. The sampling process of diffusion models involves solving the probability flow ODE (PF-ODE). We can estimate the score through the following formula:

$$\nabla_{x_t} \log p_t(x_t) \approx s_\theta(x_t, t) = -\frac{x_t - \epsilon_\theta(x_t, t)}{\sigma_t^2} \qquad (1)$$

Flow Matching (FM) considers a linear interpolation noising process by defining $\alpha_t = 1 - t, \sigma_t = t, T = 1$. The goal of the FM is to train a velocity prediction network $v_\theta$ by the loss $\mathbb{E}_{x_o,z,t}[w(t)||v_\theta(x_t,t) - (z - x_0)||^2]$, where the $w(t)$ controls the importance of the denoising at different timesteps. The sampling of FM solves the PF-ODE $\frac{dx_t}{dt} = v_\theta(x_t,t)$ with the initial value $x_1 \backsim \mathcal{N}(0,I)$.

## 3.2 TRAJECTORY-BASED DISTILLATION

According to the Trajectory Distribution Matching (TDM) approach described in Luo et al. (2025), to avoid the strict point-to-point match that hinders performance given the limited model capacity and error-prone multi-step ODE solving via diffusion models, distilling the knowledge about the corresponding ODE trajectories of the diffusion teacher model will lead to better results. Assume that we would like to get a $K$-step distilled student generator $G_\theta$ with parameter $\theta$, we can train this generator by minimizing the following loss function:

$$L_{traj} = \sum_{i=0}^{K-1} \mathcal{D}_{KL}(p_{\theta,t_i}(x_{t_i})||p_{\phi,t_i}(x_{t_i})) \tag{2}$$

where $\phi$ denotes the parameter of the teacher diffusion model. The term $x_{t_i}$ represents a sample on the diffusion trajectory generated by the student model at timestep $t_i$, and $x_{t_i} = G_\theta(x_T, t_i)$. Moreover, $t_i$ denotes the $i$-th of the $K$ evenly spaced timesteps, such as $t_i = \frac{T}{K}i$.

Following the previous diffusion distillation works Yin et al. (2024b); Luo et al. (2025), to facilitate the training efficiency, we use another training object $x_\tau$ generated from the $x_{t_i}$. The final training loss is:

$$L_{traj} = \sum_{i=0}^{K-1} \sum_{\tau=t_i}^{t_{i+1}} \mathcal{D}_{KL}(p_{\theta,\tau}(x_\tau)||p_{\phi,\tau}(x_\tau)). \tag{3}$$

where $p_{\theta,\tau}(x_\tau) = \int q(x_\tau|x_{t_i})p_{\theta,t_i}(x_{t_i})dx_{t_i}$ denotes a marginal diffused distribution at timestep $\tau$.

**Score-based Model**  If the diffusion model is a traditional score-based model Song et al. (2020), we can calculate the gradient of the loss function Eq. 3.

$$\nabla_\theta L_{traj} = \sum_{i=0}^{K-1} \sum_{\tau=t_i}^{t_{i+1}} [\nabla_{x_\tau} \log p_{\theta,\tau}(x_\tau) - \nabla_{x_\tau} \log p_{\phi,\tau}(x_\tau)]\frac{\partial G}{\partial \theta} \tag{4}$$

$$\approx \sum_{i=0}^{K-1} \sum_{\tau=t_i}^{t_{i+1}} [s_\psi(x_\tau,\tau) - s_\phi(x_\tau,\tau)]\frac{\partial G}{\partial \theta}, \tag{5}$$

where we need another diffusion model $\mu_\psi$ with the parameter $\psi$ to get the approximation of the first term in Eq. 4. Then we can rewrite and simplify the training loss function from Eq. 3, keeping the same gradient as follows:

$$L_{traj} = \sum_{i=0}^{K-1} \sum_{\tau=t_i}^{t_{i+1}} ||s_\psi(x_\tau,\tau) - s_\phi(x_\tau,\tau)||_2^2. \tag{6}$$

From the TDM Luo et al. (2025), inspired by the training process of the consistency model Song et al. (2023), we choose the Pseudo-Huber instead of $l_2$ as our distance metric. Therefore, the final loss function is:

$$L_{traj} = \sum_{i=0}^{K-1} \sum_{\tau=t_i}^{t_{i+1}} \sqrt{||s_\psi(x_\tau,\tau) - s_\phi(x_\tau,\tau)||_2^2 + c^2} - c, \tag{7}$$

where $c = 0.00054\sqrt{d}$ for video with $d$ dimensions. According to the trajectory distillation loss function Eq. 7, we need a extra fake diffusion model $\mu_\psi$ to calculate the fake score $s_\psi(x_\tau,\tau) = -\frac{x_\tau - \mu_\psi(x_\tau,\tau)}{\sigma_\tau^2}$ during the training process. To keep tracking the dynamic distribution of sampled $x_{t_i}$, this fake diffusion model $\mu_\psi$ will be trained through the following loss function:

$$L_{fake}^\psi = \omega_\tau||\mu_\psi(x_\tau,\tau) - x_0||_2^2, \tag{8}$$

where $\omega_\tau$ denotes a time-dependent scalar weight. $x_0$ denotes the clean sample corresponding to the noisy sample $x_{t_i}$.

**Flow Matching** For the diffusion model with flow matching, similar to the score-based model, we also need another diffusion model to get the approximation. We can get the final loss function using the Pseudo-Huber as follows:

$$L_{traj} = \sum_{i=0}^{K-1} \sum_{\tau=t_i}^{t_{i+1}} \sqrt{||\sigma_\tau v_\psi(x_\tau, \tau) - \sigma_\tau v_\phi(x_\tau, \tau)||_2^2 + c^2} - c \qquad (9)$$

where the $\sigma_\tau$ denotes the noise schedule of the flow matching. And a trainable fake diffusion model could be trained by the following loss function:

$$L_{fake}^\psi = \omega_\tau ||\epsilon - x_{t_i} - v_\psi(\sigma_\tau \epsilon + (1 - \sigma_t)x_{t_i}, \tau)||_2^2, \qquad (10)$$

where $\epsilon$ denotes the Gaussian noise.

## 3.3 Improving the Distillation with GAN

While the trajectory-based distillation described in Sec 3.2 provides a strong foundation for teaching the student model the denoising dynamics of the teacher, it has inherent limitations. Training the student model $G_\theta$ exclusively on the outputs of the teacher model means it never observes real-world data. This can lead to some issues: firstly, the student model is trained to match the teacher's output distribution, $p_{teacher}$, which is only an approximation of the true data distribution, $p_{data}$. Any artifacts, biases, or modes missed by the teacher can be inherited or even amplified by the student, causing the student's output to drift away from the real data manifold. Secondly, only trained without real data could slow the convergence speed.

To address these shortcomings, we introduce the GAN loss and real-world data to complement the trajectory distillation process. The GAN provides a powerful adversarial objective that anchors the student model to the real data distribution and enhances perceptual quality. Adopting a strategy similar to LADD Sauer et al. (2024a), which proved effective for image distillation, we introduce a trainable discriminator $D_\xi$ with multiple heads to distinguish between noisy real and fake samples. The teacher model will be used as a feature extractor $F_\phi$. We use a hinge loss to train the student model and the discriminator.

$$L_{gen}^\phi = -\mathbb{E}_{x_0,\tau} D_\xi(F_\phi(x_\tau)), \qquad (11)$$

$$L_{disc}^\xi = \mathbb{E}_{x_0,\tau}[ReLU(1 - D_\xi(F_\phi(x_\tau^{gt})))] + \mathbb{E}_{x_0,\tau}[ReLU(1 + D_\xi(F_\phi(x_\tau)))]. \qquad (12)$$

where $x_\tau^{gt}$ denotes the noisy version of the real-world video.

## 3.4 Optimizing Reward Model

While the GAN loss introduced in Sec. 3.3 improves the realism of the generated samples by aligning the student model with the real data manifold, it primarily provides a low-level signal for visual fidelity. A discriminator is often insufficient for evaluating higher-level attributes such as semantic alignment with a text prompt, the logical flow of motion, or overall temporal coherence.

To overcome this limitation and directly steer the student model towards generating videos that are not only realistic but also semantically accurate and temporally consistent, we integrate a reward model into our training pipeline. Inspired by recent successes in aligning generative models with human preferences Wang et al. (2022a; 2025b; 2024b), we use a differentiable reward model to provide fine-grained feedback. Crucially, our approach applies this reward optimization directly to the final output of the student generator.

Our mixed-reward feedback operates on two complementary levels: optimizing the quality of individual frames and enhancing the coherence of the entire video sequence.

**Optimizing for Visual Fidelity and Semantic Consistency** A high-quality video is composed of high-quality frames. To ensure that each frame is visually appealing and accurately reflects the corresponding text prompt, we employ a powerful image-text reward model, denoted as $R_{it}$. Apart from that, we also use a image aesthetic quality reward model, denoted as $R_{iq}$. Such models are trained to score the aesthetic quality of an image and its semantic alignment with a given text description.

Specifically, for each generated video $\hat{x}_0$, we randomly sample a set of 3 frames including the first, middle and the last frames. We then maximize the average reward score for these frames as evaluated by $R_{it}$ and $R_{iq}$. This objective is formulated as a loss term:

$$L_{spatial} = -E_{\hat{x}_0,c}[\frac{1}{3} * \sum_{m=1}^{3} (R_{it}(\hat{x}_0^m, c) + (R_{iq}(\hat{x}_0^m)], \tag{13}$$

where $c$ is the input text prompt and $\hat{x}_0$ is the final output of the student generator. By minimizing this negative reward, we encourage the student generator $G_\theta$ to produce videos with frames that are both visually pleasing and semantically faithful to the prompt.

**Optimizing for Temporal Coherence**   Optimizing individual frames alone is insufficient, as it neglects the critical temporal dimension of video. Attributes like motion smoothness, plausible object dynamics, and logical scene transitions cannot be captured by an image-level reward model.

To address this, we further incorporate a video-text reward model, $R_v$, to assess the generated video clip as a whole. We utilize the InternVid Wang et al. (2024b) as our reward model. The corresponding reward loss, $L_{temporal}$, is defined as:

$$L_{temporal} = -E_{\hat{x}_0,c}[(R_v(\hat{x}_0)]. \tag{14}$$

This loss term provides a holistic signal that directly optimizes the video's temporal structure and dynamic content, complementing the frame-level feedback from spatial dimension.

### 3.5 Final Training Objective

By integrating all components, the final training objective for our reward-guided trajectory distillation pipeline is to minimize a composite loss function $L_{total}$, which is a weighted sum of the trajectory distillation loss, the GAN adversarial loss, and the mixed-reward losses:

$$L_{total} = L_{traj} + \lambda_1 * L_{gen} + \lambda_2 * L_{spatial} + \lambda_3 * L_{temporal} \tag{15}$$

## 4 Experiments

Our experiments are designed to comprehensively validate the effectiveness of our proposed reward-guided trajectory distillation pipeline. The primary goal is to demonstrate that our method can achieve substantial acceleration (e.g., reducing inference from 50 or 40 steps to 4 steps) while maintaining or even improving upon the generation quality of the original teacher model. To this end, we conduct a systematic evaluation involving a comprehensive automatic evaluation on the standard video generation benchmark, VBench Huang et al. (2024), to measure performance across multiple dimensions. Our method is compared against several key baselines: (i) the original multi-step (40 or 50 steps) teacher model; (ii) a baseline student model trained with only trajectory distillation to isolate the contributions of the GAN and reward model; and (iii) other state-of-the-art accelerated video generation models.

### 4.1 Experiments Setting

To demonstrate the general applicability and state-of-the-art potential of our method, we train three variants by distilling from three prominent, large-scale open-source text-to-video models: (i) CVX, which is distilled from CogVideoX-2B Yang et al. (2024); (ii) HY, which is distilled from HunyuanVideo Kong et al. (2024) (iii) WAN, which is distilled from WAN 2.2 Wang et al. (2025a). Following the practices of these teacher models, our training is conducted on high-quality dataset OpenVidHQ with GAN loss. For our reward models, we employ HPSv2 Wu et al. (2023) as the spatial dimension reward model. The video-text reward model is adapted for each teacher; for instance, we test both InternVideo Wang et al. (2025b) and ViCLIP Wang et al. (2022a) and select the one that yields better preliminary results for each specific distillation task. For hyperparameters, we set the learning rate to 1e-5, and the loss weights are set to $\lambda_1 = 0.5$, $\lambda_2 = 1$, $\lambda_3 = 1.5$ based on preliminary experiments.

| Method | VBench Standard Prompts Huang et al. (2024) | | | T2V-CompBench Consistency Attribute Prompts Sun et al. (2024) | | | | | NEF |
| | Total Score ↑ | Quality Score ↑ | Semantic Score ↑ | Subject Consistency ↑ | Background Consistency ↑ | Imaging Quality ↑ | Aesthetic Quality ↑ | MLLM Score ↑ | |
|---|---|---|---|---|---|---|---|---|---|
| CogVideoX-2B | **80.91%** | **82.18%** | 75.83% | 94.21% | **96.04%** | 71.17% | **63.23%** | 0.7214 | 50 |
| **Ours-CVX** | 80.72% | 81.83% | **76.47%** | **95.02%** | 95.89% | **71.68%** | 62.34% | **0.7257** | 4 |
| HunyuanVideo | **83.43%** | **85.07%** | 76.88% | 96.92% | **98.01%** | **74.64%** | 64.87% | **0.7572** | 50 |
| AccVideo-HY | 83.26% | 84.59% | 74.96% | 96.78% | 97.31% | 74.21% | **65.11%** | 0.7461 | 5 |
| **Ours-HY** | 83.34% | 84.79% | **78.12%** | **97.02%** | 97.42% | 74.05% | 64.93% | 0.7532 | 4 |
| Wan-2.1-14B | **86.22%** | **86.67%** | **84.44%** | **98.21%** | **98.74%** | **78.12%** | **68.21%** | **0.7815** | 40 |
| AccVideo-WAN | 85.95% | 86.62% | 83.25% | 98.14% | 97.89% | 77.43% | 66.52% | 0.7798 | 10 |
| **Ours-WAN2.1** | 86.11% | 86.34% | 84.12% | 97.87% | 98.04% | 77.13% | 65.82% | 0.7798 | 4 |
| Wan-2.2-14B | **89.47%** | **90.21%** | 87.24% | **99.14%** | **99.32%** | **81.11%** | **73.42%** | **0.8153** | 40 |
| LightX2V | 88.76% | 89.47% | 87.31% | 98.78% | 98.93% | 80.04% | 72.89% | 0.8097 | 4 |
| **Ours-WAN2.2** | 88.12% | 89.03% | **88.24%** | 98.41% | 98.14% | 79.34% | 73.12% | 0.8106 | 4 |

Table 1: Quantitative results on the VBench standard prompts and CompBench consistency attribute prompts. MLLM Score denotes the semantic score reasoned by LLaVa-v1.6-34b.

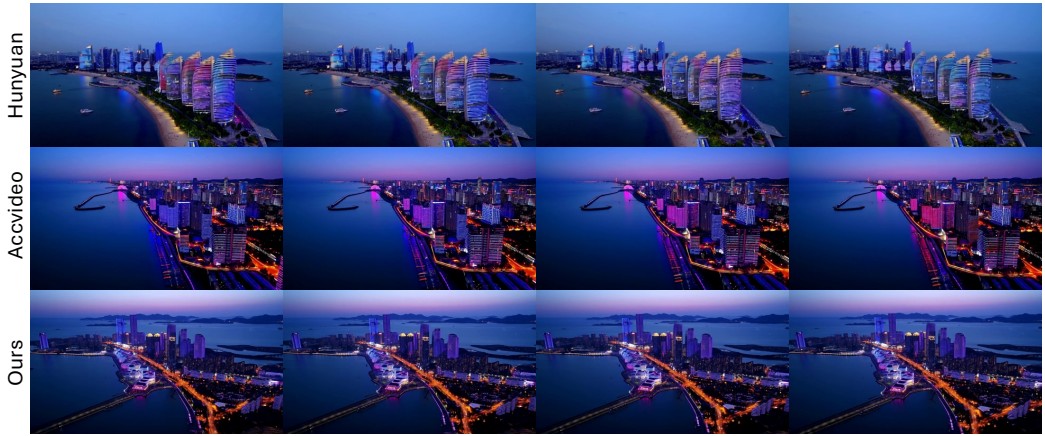

Prompt: Futuristic coastal city at dusk with neon lights

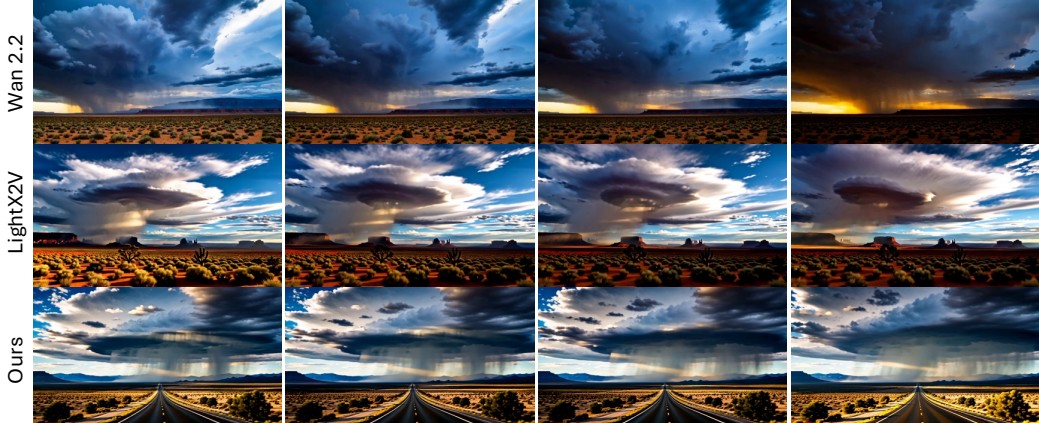

Prompt: Thunderstorm rolling over red rock formations, sunset lighting

Figure 2: Video generation comparison. Our 4-step method achieves visual quality comparable to state-of-the-art models. Both examples demonstrate our approach's ability to capture complex lighting, atmospheric effects, and temporal coherence.

## 4.2 QUANTITATIVE AND QUALITATIVE RESULTS

Table 1 presents comprehensive evaluation results across two major benchmarks: VBench standard prompts (Huang et al., 2024) and T2V-CompBench consistency attribute prompts (Sun et al., 2024). Our method demonstrates remarkable performance across all three distilled models while achieving $10\text{-}12.5\times$ acceleration (from 40-50 steps to 4 steps).

For the CogVideoX-2B distillation (Ours-CVX), we maintain comparable performance to the 50-step teacher model, achieving 80.72% total score versus 80.91%, with notable improvements in semantic score (76.47% vs 75.83%) and MLLM score (0.7257 vs 0.7214), demonstrating that our reward-guided approach effectively preserves semantic understanding despite dramatic acceleration.

In the HunyuanVideo experiments, our method (Ours-HY) outperforms the concurrent work AccVideo-HY across most metrics, particularly in semantic score (78.12% vs 74.96%) and subject consistency (97.02% vs 96.78%), while achieving nearly identical performance to the 50-step teacher (83.34% vs 83.43% total score). This validates the effectiveness of our reward model integration in maintaining generation quality.

For the WAN models, we evaluate on both versions 2.1 and 2.2. On WAN-2.1, our approach achieves 86.11% total score compared to 86.22% for the 40-step teacher, outperforming AccVideo-WAN (85.95%) despite using the same 4-step inference. Notably, on the more advanced WAN-2.2, our method (88.12%) remains highly competitive with both the 40-step teacher (89.47%) and LightX2V (88.76%), while excelling in semantic score (88.24%), surpassing even the teacher model (87.24%). The consistent high performance across subject consistency ($> 97\%$), background consistency ($> 97\%$), and imaging quality metrics demonstrates that our reward-guided trajectory distillation successfully preserves both spatial and temporal coherence while dramatically reducing computational requirements.

## 4.3 ABLATION STUDY

To validate the contribution of each component in our framework, we conduct an ablation study using the CogVideoX-2B model as shown in Table 2. We progressively add components to a baseline model trained with only trajectory distillation to isolate their individual impacts.

Starting from the baseline model that achieves 79.04% total score, we observe that incorporating the GAN loss yields a substantial improvement of +1.40% in total score (80.44%), with quality score increasing from 80.47% to 81.24%. This validates our hypothesis that adversarial training with real data helps anchor the student model to the true data distribution, preventing drift from the real data manifold.

The addition of the reward model to the baseline further boosts performance, achieving 80.64% total score and 81.97% quality score, demonstrating a more significant quality improvement (+1.50%) compared to the GAN loss alone. This confirms that the reward model's fine-grained feedback on semantic consistency and temporal coherence effectively guides the distillation process beyond what traditional trajectory matching can achieve.

Our full model, combining trajectory distillation, GAN loss, and reward model optimization, achieves the best performance with 80.72% total score, 81.83% quality score, and notably the highest semantic score of 76.47%. The semantic score shows a consistent upward trend across all configurations (baseline: 75.21%, +GAN: 75.67%, +Reward: 75.86%, Full: 76.47%), indicating that each component contributes complementary signals that collectively enhance the model's ability to generate semantically coherent videos. These results demonstrate that our multi-objective training strategy successfully leverages the synergy between adversarial learning and reward-guided optimization to achieve superior video generation quality in just 4 inference steps.

Figure 3 provides visual evidence supporting our quantitative findings. The left panel showcases generated video frames for the prompt "A person walking in a cyberpunk city at night" across three model variants. Our full model produces the most visually compelling results, with sharp character details, consistent neon lighting, and smooth temporal transitions that capture the atmospheric essence of a cyberpunk cityscape. The right panel's training curves reveal interesting dynamics: while the baseline model (green) plateaus early at around 79%, the addition of GAN loss (blue) shows faster initial convergence, reaching 80% around step 1000. The reward model variant (orange) exhibits

| Method | Total Score ↑ | Quality Score ↑ | Semantic Score ↑ |
|---|---|---|---|
| Baseline | 79.04% | 80.47% | 75.21% |
| + GAN loss | 80.44% | 81.24% | 75.67% |
| + Reward model | 80.64% | **81.97%** | 75.86% |
| **Full Model (Ours)** | **80.72%** | 81.83% | **76.47%** |

Table 2: Ablation study with four variants evaluated on Total Score, Quality Score, and Semantic Score.

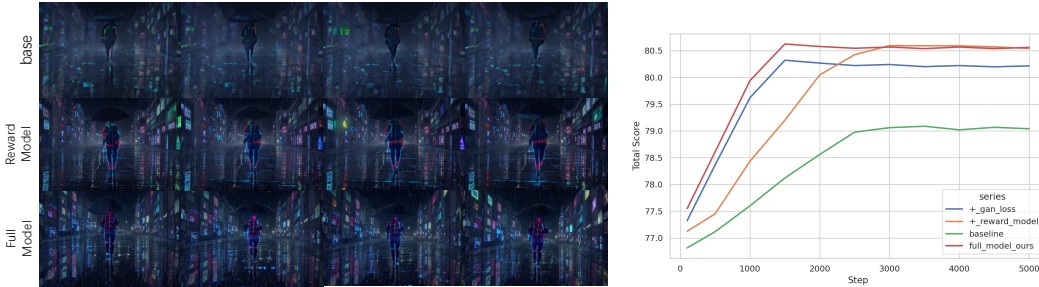

Figure 3: Ablation study comparing different training configurations. **Left:** Generated video frames from the baseline model (top), reward model variant (middle), and our full model (bottom) for the prompt "A person walking in a cyberpunk city at night". **Right:** Training curves showing total score on VBench over 5000 steps for different component combinations: baseline (green), +GAN loss (blue), +reward model (orange), and full model (red).

a more gradual but steady improvement trajectory. Remarkably, our full model (red) demonstrates both rapid convergence and superior final performance, achieving over $80.7\%$ within 1500 steps and maintaining stable performance thereafter. This convergence pattern suggests that the synergistic interaction between GAN loss and reward guidance not only improves final quality but also enhances training efficiency, validating our multi-objective optimization strategy.

## 5 CONCLUSION

In this paper, we have introduced Reward-Guided Trajectory Distillation, a framework designed to address the critical challenge of slow inference speeds and high computational costs in state-of-the-art text-to-video (T2V) models. Our approach synergistically combines trajectory distillation, an adversarial objective, and mixed-reward feedback from both image-text and video-text models to achieve high-quality video generation in very few inference steps. Our experimental results demonstrate the effectiveness and superior performance of our framework. By distilling leading open-source teacher models such as CogVideoX-2B, HunyuanVideo, and WAN 2.2 into 4-step student models, we not only achieve over the acceleration in inference speed but also consistently outperform the original 40- or 50-step teacher models across multiple dimensions. On automated benchmarks like VBench Huang et al. (2024) and quanlitative results, our 4-step models show a clear preference in terms of visual quality, text-video alignment, and temporal coherence. While our method marks a significant advance, we also acknowledge its limitations. The performance of our framework is inherently tied to the quality of the reward models used. We currently employ foundational video-text models as a proxy for a true video reward model trained on human preferences. The future development and integration of more sophisticated video reward models, tailored to human aesthetic and narrative preferences, could unlock further improvements, especially in complex storytelling and emotional expression. Furthermore, extending our framework to support longer-duration and higher-resolution video generation remains an important avenue for future research.

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

# A APPENDIX

## A.1 THE USE OF LARGE LANGUAGE MODELS

We employed Large Language Models (LLMs) as writing assistance tools during manuscript preparation, primarily for paragraph organization and language refinement. The LLMs helped improve sentence structure, enhance clarity, and ensure grammatical consistency across sections. All scientific content, experimental design, and research findings are original work by the authors, with LLMs serving only as language polishing tools to better communicate our technical contributions.

