# OpenReview forum: "Reward-Guided Trajectory Distillation for Accelerated Diffusion-Based Video Generation"
_ICLR.cc/2026/Conference — ICLR 2026 Conference Withdrawn Submission_

### Official Review · Reviewer_H9VB · 2025-10-19

**Soundness:** 2
**Presentation:** 2
**Contribution:** 3
**Rating:** 2
**Confidence:** 4

**Summary:**

This paper introduces a reward-guided trajectory distillation framework to accelerate diffusion-based video generation. This method is a combination of previous TDM on image generation and reward fine-tuning.
The method distills a 40–50-step teacher diffusion model into a 4-step student model by matching the trajectory distribution between teacher and student while integrating GAN loss and multi-level reward guidance.
The approach is validated on CogVideoX, HunyuanVideo, and WAN models, achieving 10–12× faster inference while maintaining competitive benchmark metrics (e.g., VBench and CompBench).

**Strengths:**

1. This work combines trajectory distillation, adversarial training, and reinforcement-style reward optimization in a coherent pipeline that addresses both efficiency and quality.
2. The method demonstrate strong empirical results, consistently matches or outperforms 40–50-step teachers with only 4 steps, demonstrating significant acceleration (10×–12.5×) and robustness across diverse architectures.

**Weaknesses:**

1. Limited novelty: While the combination of trajectory distillation and reward guidance is effective, the individual components (TDM, GAN, RLHF-like rewards) and combination of distillation and GAN or reward are already largely adaptations of existing ideas.
2. Although inference is accelerated, training involves multiple auxiliary networks (fake score model, discriminator, reward models), which could offset efficiency gains.
3. Misclaim and errors:

3.1. "However, applying such techniques to video generation is still to be explored." this is not true, diffusion distillation has been largely used for video diffusion models already.

3.2 in the caption of fig1, what is $t_i$, $x_\tau$ and $\hat{x}_0$, no correspondence can be found in the figure but in the main paper. please address this inconsistency.

3.3 line 157 and line 164, should be $x_0$ rather than $x_o$

3.4 line 227, the noise is $\epsilon$ but in the previous part the noise is $z$, why is that?

3.5 $\hat{x}_t^m$ is undefined in eq(13)

3.6 how is $c$ used in eq(14)

4. equation (5) and (6) are different, the first optimize the KL while the second optimize the fisher divergence.

**Questions:**

1. which sampler is used for the generator during distillation?
2. in line 189, what do you mean by 'If the diffusion model is a traditional score-based model'? aren't they equivalent

---

### Official Review · Reviewer_U2pM · 2025-10-28

**Soundness:** 3
**Presentation:** 2
**Contribution:** 2
**Rating:** 4
**Confidence:** 4

**Summary:**

This paper tackles the speed bottleneck in diffusion-based video generation by proposing a novel training pipeline. It distills a few-step video diffusion models into a fast few-step generator. The core idea is trajectory distillation. The authors train a student video generator to match the trajectory of a pre-trained diffusion teacher that effectively compresses the iterative denoising process from dozens of steps down to only 4 inference steps without significant quality loss. To preserve and even enhance generation quality during this aggressive distillation, the method integrates two additional components into the training. First, an GAN loss to enforce realism and sharpness in frames and second, a multi-faceted reward model that guides the generator toward better semantic alignment, visual fidelity, and temporal coherence. The reward model provides differentiable feedback on both image-level and video-level attributes, for example, scoring how well each frame matches the text prompt and how consistent the whole clip is over time. By combining trajectory matching with these guidance signals, the distilled student is trained to rapidly generate high-quality videos that are both faithful to the input text (when given) and temporally smooth. The paper’s contributions include introducing this reward-guided distillation framework, a reward signal composed of image-text and video-text evaluators, and evaluations demonstrating 10-12x faster video generation with comparable output quality to the original diffusion models.

**Strengths:**

1. The paper combines diffusion model distillation with a learned reward-guided objective in the video domain. The author(s) blend trajectory distillation (matching the teacher's denoising path) with reinforcement learning-style feedback. The multi-level reward signal (image-level aesthetic + semantic rewards, plus a video-level coherence reward) is an original contribution that goes beyond prior distillation works, which typically optimize only for pixel or feature differences. By directly optimizing for high-level criteria (e.g. text alignment and temporal consistency), the approach addresses quality aspects that normal distillation might miss.

2. The paper distilled three different large teacher models, i.e., CogVideoX-2B, HunyuanVideo, and WAN 2.2 demonstrating the framework's generality to multiple architectures. The results are measured on VBench, a standard automated video generation benchmark, covering aspects like visual quality, text relevance, and temporal smoothness. The paper also compares against other acceleration methods (e.g. prior consistency distillation and adversarial distillation baselines). The distilled models are shown to be competitive or better in all categories. The author(s) include ablation studies that isolate the effect of each component i.e., trajectory distillation baseline vs. adding GAN loss vs. adding the reward optimization.

3. The method achieves nearly 10x-12x speed-up in video sampling. The experiments show that these 4-step models match or even outperform the original long-run diffusion models on multiple metrics. This seems like a convincing result, typically one expects some degradation when reducing steps. For example, when distilling the large Hunyuan Video model, the 4-step student outperforms the 40-step teacher in semantic alignment and subject consistency scores.

**Weaknesses:**

The paper weaknesses revolve around the dependence on external reward heuristics and evaluation gap. The paper in its current form might be slightly optimistic about the method's generality and impact. Concretely comments are detailed below:

1. A core component of this approach is the use of external reward models. The performance of the distilled generator is greatly tied to the quality and biases of these reward models. This raises concerns about generality and true alignment.  The reward models used are proxies for human judgment, not actual human preference models. If these models have blind spots or biases, the student generator will optimize toward those, potentially leading to misguided improvements. The paper does not report any human evaluation, so it's unclear if the reward-defined improvements correspond to genuine perceptual enhancements. The authors themselves acknowledge that more sophisticated video reward models trained on human preferences are needed and that they currently rely on foundational models as a proxy. Without a human study or more diverse evaluation, there's a risk that some methods are overfit to the reward models' criteria.

2. The paper mentions extending to longer/higher-resolution videos as future work. Still, it remains unclear whether the distilled 4-step model can maintain coherence over, say, tens of seconds or complex multi-scene stories. The reward model's temporal coherence signal (from a video-text model) might not be sufficient for very long sequences that involve more complicated scene changes or long-range dependencies. Also, the trajectory distillation technique might need modification for substantially longer generation horizons; for example, distilling 1000 diffusion steps to 40 steps for a minute-long video poses new challenges for preserving fidelity. Thus, the generality of the proposed pipeline to videos slightly longer than the training set (OpenVidHQ) is unproven, and no analysis is provided on how the number of steps might scale for much such longer videos.

3. The method assumes access to a large pre-trained diffusion teacher (e.g., 2-billion-parameter CogVideoX) and then performs distillation on a possibly different dataset (the paper used OpenVidHQ for training students, which may or may not be the original data of the teachers). If the training (distillation) dataset differs from the teacher's training data, there is a mismatch that the paper alludes to as a challenge but does not deeply examine. In practice, the student could be learning beyond the teacher if the new dataset includes new scenes or if the student is effectively fine-tuned to a specific domain. This complicates the interpretation of results. For instance, the paper reports that the student outperforms the teacher model on some metrics, but is this because the distillation process inherently improves the model, or simply because the student was trained on a higher-quality or more domain-focused dataset? Suppose the teacher was not fine-tuned on OpenVidHQ (assuming the teacher models were originally trained on broader web data). In such a case, the student's superior performance may partly stem from exposure to a cleaner and more focused dataset. The paper do not discuss such cases. Failing to fine-tune or evaluate the teacher under the same data conditions could make the comparison less fair. In terms of generality, the need for a specific large teacher per domain means the method is not a stand-alone generator; it accelerates existing models but doesn't inherently improve diversity beyond what the teacher could generate. If one wanted to apply this to a new domain (say, medical video or animation), one must first have a diffusion model for that domain. The paper's contribution is thus somewhat tied to the availability of influential teachers, and it doesn't explore distilling smaller or different-architecture students (e.g., could the student model be not just faster but also lighter in parameters?). This reliance might limit the framework's applicability when such a teacher model is not readily available or when memory constraints call for model compression in addition to step reduction.

4. The paper mentions using an image-text reward (likely CLIP-based) and an image aesthetic model, summed together, as well as a video-text model (InternVideo or ViCLIP). However, it provides few details about these models (e.g., architectures or training data) aside from citations and how they are combined. For example, HPSv2 (Wu et al., 2023) is referenced as the spatial reward; presumably, this is a model that outputs a single score given an image and a text prompt. Two separate networks or two HPSv2 heads? It's unclear. Also, the video reward model is "adapted for each teacher", does this mean it was fine-tuned on videos similar to the teacher's content, or selected (InternVideo vs ViCLIP) based on which worked better? The lack of explicit reward model fine-tuning to the specific domain of the videos is a mild concern. If the reward model doesn't perfectly align with the data distribution, the optimization might be suboptimal. (The authors acknowledge this and suggest future work on better reward models.)

**Questions:**

Q1. The author(s) optimize the generator using learned reward models. How do you ensure that optimizing these rewards genuinely improves human-perceived video quality and relevance, rather than overfitting to the quirks of the reward models? For instance, did you observe any cases of the model exploiting the reward (e.g., producing overly saturated or unnatural frames that trick the aesthetic model into a high score)? Conducting a small human evaluation or showing side-by-side outputs would help verify that the reward model's notion of quality aligns with people's preferences. In the absence of that, can you provide qualitative insights or examples that demonstrate the improvements are subjectively meaningful and not just numeric boosts in proxy metrics?

Q2. In your training, you sample only three frames from each video to compute the image-level reward loss. Could this sparse sampling miss issues in intermediate frames (for example, a brief glitch or a momentary lapse in fidelity that doesn't occur precisely at the midpoint)? Did you experimented with using more frames or even all frames for the image-level reward? If the model were to generate longer videos, would you keep using just the first/middle/last frame for reward feedback, or might a more extensive frame sampling be needed?

Q3. The student model is distilled using the OpenVidHQ, which differs from the data the teachers were trained on. Can we fine-tune the teacher models this dataset, or use the same training data for them to ensure a more useful comparison?

Is it possible that the student's improved performance is partly due to training on a different or higher-quality data distribution than the teacher had seen? Essentially, how to disentangle the benefits of the distillation algorithm from the potential benefits of additional training data or GAN fine-tuning that the teacher did not receive? This is important for evaluating whether the acceleration comes at any hidden cost (e.g., narrower generalization) or whether, conversely, the teacher could also improve with the same training refinement.

Q4. What challenges do you anticipate in extending your reward-guided distillation to generate longer videos or higher-resolution content? For example, would the current reward models scale to assessing videos that are, say, 30 seconds long or 1024×768 resolution? Training on longer sequences might also require distilling a longer diffusion chain.

---

### Official Review · Reviewer_4JZ6 · 2025-10-30

**Soundness:** 2
**Presentation:** 2
**Contribution:** 2
**Rating:** 2
**Confidence:** 4

**Summary:**

The paper proposes Reward-Guided Trajectory Distillation (RGTD) to compress multi-step text-to-video diffusion teachers into a 4-step student. The student is trained with (i) trajectory distribution matching using a trainable “fake score/velocity” model and a Pseudo-Huber metric, (ii) a GAN head that uses frozen teacher features to anchor realism, and (iii) mixed rewards: frame-level (image–text + aesthetics) and video-level (video–text) signals. On VBench and T2V-CompBench, 4-step students distilled from CogVideoX-2B, HunyuanVideo, and WAN-2.1/2.2 roughly match their 40–50-step teachers. Ablations attribute complementary benefits to the GAN and reward components.

**Strengths:**

* Practical acceleration: Cuts inference from 40–50 steps to 4 while largely preserving automatic metrics—valuable for deployment.
* Coherent objective: Clean integration of trajectory matching, adversarial training, and multi-granular rewards; formulation is explicit and reproducible.
* Model-agnostic evidence: Results across three major teachers (CogVideoX, Hunyuan, WAN-2.2/2.1) suggest portability of the pipeline.
* Ablations: Show that GAN improves perceptual quality and rewards help semantics, aligning with design intent.

**Weaknesses:**

* Limited novelty/Delta: Largely combines known ideas—trajectory/consistency distillation + GAN loss + RL/reward tuning; contribution feels incremental without sharper analysis vs. concurrent video accelerators (e.g., AccVideo/LightX2V/APT).
* Modest gains: Improvements over teachers/baselines are small on many metrics; claims of superiority are not consistently significant.
* Qualitative evidence is thin: Only a couple of static examples; no live demo/gallery is linked or provided in supplementary materials.
* Reward design risk: Frame rewards use only 3 frames (first/middle/last), potentially biasing toward static scenes; no analysis of reward hacking.
* Missing sensitivity: (\lambda) weights are fixed ((\lambda_1=0.5,\lambda_2=1,\lambda_3=1.5)) with no coefficient study

**Questions:**

Please refer to weaknesses

---

### Note · Authors · 2025-11-14

I have read and agree with the venue's withdrawal policy on behalf of myself and my co-authors.